# Integrative Bioinformatics–Gene Network Approach Reveals Linkage between Estrogenic Endocrine Disruptors and Vascular Remodeling in Peripheral Arterial Disease

**DOI:** 10.3390/ijms25084502

**Published:** 2024-04-19

**Authors:** Vincent Avecilla, Mayur Doke, Madhumita Das, Oscar Alcazar, Sandeep Appunni, Arthur Rech Tondin, Brandon Watts, Venkataraghavan Ramamoorthy, Muni Rubens, Jayanta Kumar Das

**Affiliations:** 1Robert Stempel College of Public Health & Social Work, Florida International University, Miami, FL 33199, USA; vavec001@fiu.edu; 2Diabetes Research Institute, University of Miami, Miami, FL 33136, USA; mdoke001@fiu.edu (M.D.); o.alcazar@med.miami.edu (O.A.); axr1957@miami.edu (A.R.T.); bhw23@miami.edu (B.W.); 3Department of Biology, Miami Dade College, Miami, FL 33132, USA; mdas@mdc.edu; 4Department of Biochemistry, Government Medical College, Kozhikode 673008, Kerala, India; sandeepappunni@gmail.com; 5Baptist Health South Florida, Miami Gardens, FL 33176, USA; drvenky37@gmail.com (V.R.); mrube001@fiu.edu (M.R.); 6Department of Health and Natural Sciences, Florida Memorial University, Miami Gardens, FL 33054, USA

**Keywords:** estrogenic endocrine disruptors, gene–environment interaction, gene network analysis, peripheral arterial disease, vascular remodeling

## Abstract

Vascular diseases, including peripheral arterial disease (PAD), pulmonary arterial hypertension, and atherosclerosis, significantly impact global health due to their intricate relationship with vascular remodeling. This process, characterized by structural alterations in resistance vessels, is a hallmark of heightened vascular resistance seen in these disorders. The influence of environmental estrogenic endocrine disruptors (EEDs) on the vasculature suggests a potential exacerbation of these alterations. Our study employs an integrative approach, combining data mining with bioinformatics, to unravel the interactions between EEDs and vascular remodeling genes in the context of PAD. We explore the molecular dynamics by which EED exposure may alter vascular function in PAD patients. The investigation highlights the profound effect of EEDs on pivotal genes such as ID3, LY6E, FOS, PTP4A1, NAMPT, GADD45A, PDGF-BB, and NFKB, all of which play significant roles in PAD pathophysiology. The insights gained from our study enhance the understanding of genomic alterations induced by EEDs in vascular remodeling processes. Such knowledge is invaluable for developing strategies to prevent and manage vascular diseases, potentially mitigating the impact of harmful environmental pollutants like EEDs on conditions such as PAD.

## 1. Introduction

Vascular remodeling is a sophisticated physiological process that involves changes to blood artery structure. These changes frequently take place in response to a variety of stimuli, including changes in blood flow, mechanical stress, or pathogenic causes. It includes changes to vessel wall thickness, diameter, and composition that eventually impact vascular tone, compliance, and resistance [1]. Peripheral arterial disease (PAD), which is characterized by atherosclerotic plaques that have restricted the arteries, is a serious cardiovascular condition that has an enormosus effect on world health [2]. In addition to impairing wound healing and causing claudication, discomfort, and restricted blood flow to the limbs, PAD also acts as a standalone predictor of unfavorable cardiovascular events. Given its high incidence, affecting around 202 million people worldwide, PAD is linked to significant morbidity, mortality, and healthcare expenditures. It is essential to comprehend the complex mechanisms behind vascular remodeling in the setting of PAD in order to create efficient preventive and treatment measures [2,3,4,5]. Therefore, it is essential to develop technologies for the early detection of PAD as it allows for timely interventions that can significantly decrease the risk of heart attacks and strokes, thus preventing major adverse cardiovascular events [6].

Environmental contaminants known as estrogenic endocrine disruptors (EEDs) are known for their capacity to disrupt endocrine systems and imitate the effects of natural estrogens in the body. These substances are widely found in a variety of products, such as plastics, industrial chemicals, and pesticides. They are known to have the ability to disturb hormonal balance, which can result in a variety of harmful health effects. EEDs have drawn a lot of study attention because of their links to biological consequences that are diverse and have implications for vascular health [7,8,9,10]. Due to their status as persistent organic pollutants, EEDs have the propensity to amass within organs with lipid-rich compositions. This accumulation can lead to direct exposure of endothelial cells lining the innermost layer of pulmonary arteries and arterioles [10]. Consequently, this exposure has the potential to exacerbate inflammation during instances of vascular injury. According to studies, EED exposure has been linked to endothelial dysfunction, oxidative stress, inflammation, and changes in vascular tone. For instance, studies have shown that some EEDs might influence nitric oxide synthesis and hinder appropriate vasodilation, which can result in vascular dysfunction. In a study by Dubey et al., endothelial dysfunction was linked to exposure to the EED bisphenol A (BPA) in animals and humans [11]. Similarly, another research group revealed a connection between EED exposure and an elevated risk of cardiovascular illnesses [12]. These results highlight the potential for EEDs to influence vascular health via pathways that call for additional research, stressing the importance of comprehending their impacts concerning conditions like PAD.

At the nexus of biology and computational science, bioinformatics plays a key role in enabling the extraction of significant insights from massive biological data. We used the bioinformatics data mining approach to understand the complex connections between EEDs and vascular remodeling in PAD. To clarify the relationships between genes and their functions in the context of EED exposure and its impact on vascular health, the approach of gene network analysis is specifically used [13]. A gene network analysis examines how genes are related in biological systems. It entails building networks with nodes that stand in for genes and edges that indicate interactions between them. Researchers can acquire a comprehensive grasp of how genes interact in intricate pathways and regulatory mechanisms by unraveling these networks. The complex methods by which EEDs affect the genes involved in vascular remodeling and contribute to the onset of PAD are uncovered in this study using gene network analysis. This method makes it easier to identify important genes, their regulatory connections, and probable pathways that EEDs may disrupt, providing insights into the molecular causes of vascular disorders [13,14].

This study combines data from several sources to build a thorough framework for analysis. These sources might include information on genetic interactions, gene expression profiles, and pertinent biological annotations. The integration of these many data sources permits a thorough investigation of the molecular environment influenced by EEDs. The study team creates a composite dataset through data integration and processing that captures the complex interactions between genes, their expressions, and their interactions in the setting of EED exposure. The resulting gene network analysis is built on this integrated dataset, which offers a thorough understanding of the molecular dynamics relating to EEDs and vascular remodeling in PAD. Combining bioinformatics methods with various data sources enables a more comprehensive comprehension of the intricate biological mechanisms behind vascular disorders and environmental influences [14,15].

The approach used in this study’s methodology, known as gene network analysis, reveals the deep relationships between the genes affected by EEDs and the process of vascular remodeling in PAD. In order to find gene relationships, understand pathways, and identify putative regulatory mechanisms impacted by EED exposure, the investigation makes use of cutting-edge computer tools. These techniques combine information on gene expression, genetic relationships, and other pertinent factors to build elaborate networks that demonstrate the dynamic interactions between genes in the context of EED-mediated vascular alterations. An important result of this investigation is the identification of important genes. Notably, genes are identified that have a big impact on how EEDs and vascular remodeling interact in PAD. These genes were chosen based on their essential roles in vascular biology, frequently taking part in vital regulatory processes or assisting in known disease-related pathways. Researchers can learn more about the molecular causes of vascular disorders and how EED exposure can accelerate the process of vascular remodeling by examining these genes, which advances our understanding of the environmental factors that contribute to these conditions.

The complicated web of connections that connects EED exposure to vascular remodeling can be better understood mechanistically thanks to gene network analysis. The identified genes and their interactions map complex biological networks that provide insight into how EEDs affect vascular alterations in PAD. By identifying potential targets that may be altered to lessen the effect of EEDs on vascular health, this approach may provide novel therapeutic options. Beyond the parameters of this particular investigation, the findings of this study have significance for vascular biology and environmental health. This study contributes important knowledge to the field by illuminating the gene networks underpinning EED-mediated vascular remodeling and offering prospective entry points for therapies that target the environmental causes of vascular disorders.

## 2. Results

### 2.1. Selection and Data Integration

During the early stage of our study, we conducted a thorough selection and integration procedure to establish the foundation for our complete analysis. The process entailed a thorough examination of relevant scientific literature, employing PubMed and other scientific databases, to pinpoint estrogens and EEDs that have a confirmed connection to cardiovascular and vascular illnesses. The results of this analysis, presented in Table 1, have established a fundamental comprehension of the effects of these substances on health. Afterward, we utilized the NCBI GEO database, which is a comprehensive collection of functional genomics data, to advance our inquiry. In this study, our main focus was on dataset GSE27034, which provided useful information about genes that were DEGs and the interactions between genes in a group with PAD. The dataset, described in Table 2, consisted of expression profiling by array from 19 PAD patient samples and 18 controls. This dataset serves as a strong foundation for our gene interaction study. The incorporation of these varied data sources facilitated a comprehensive method for comprehending the intricate interaction between ambient estrogens, EEDs, and genetic variables in the development of vascular illnesses, specifically PAD. The extensive data integration not only enhanced our research but also guaranteed the strength and relevance of our findings within the framework of current scientific knowledge.

### 2.2. Estrogen-Interacting Genes and EED-Interacting Genes

Expanding upon the initial data integration, our work advanced to a thorough analysis of the interaction between environmental estrogens and EEDs with the vascular system. We conducted a comprehensive investigation of CTD, where we methodically identified genes that interact with significant environmental chemicals such as polychlorinated biphenyls (PCBs), BPA, phthalates, and estrogenic compounds such as DES and E2. The CTD’s extensive curation, utilizing a wide range of published literature, played a crucial role in identifying genes associated with vascular remodeling and PAD. The process of integrating data in this phase was thorough and precise, as it involved carefully comparing our findings from the GEO dataset with those in the CTD. This ensured a comprehensive understanding of the relationships between genes and the environment. Through the comparison of these gene lists with established pathways and disease connections, we have obtained an unparalleled understanding of the molecular processes by which EEDs impact vascular health. This comprehensive approach not only enhanced our comprehension of the interaction between genes and the environment in vascular disorders but also paved the way for a more focused exploration of the particular genetic pathways and networks influenced by exposure to EED in the context of PAD. The comprehensive integration and analysis of data highlighted the intricate nature of gene–environment interactions and their importance in the development of vascular disorders.

### 2.3. Vascular Remodeling and Peripheral Arterial Disease Genes

The thorough examination of genes that interact with EEDs has resulted in an intricate network of intersections, as seen in Figure 1. The Venn diagrams depict the intersecting gene interactions among four categories of EEDs: diethylstilbestrol, E2, BPA, and phthalates. Additionally, each of these groupings is compared to PCBs. The analysis reveals that diethylstilbestrol exhibits interactions with a total of 1400 genes. Among these, 335 genes are exclusively associated with DES, while 1065 genes overlap with those that also interact with E2. In total, E2 interacts with 6894 genes (Figure 2A) (Appendix A). The presence of an overlap between DES and E2 suggests that they share a route or have comparable methods of action. This is supported by the fact that 5829 genes do not interact with DES. This indicates that E2 has a wider range of influence on the genetic landscape (Figure 2A). BPA, phthalates, and PCBs exhibit a complex network of gene interactions. BPA interacts with a total of 20,515 genes. Out of these, 11,024 genes are specifically affected by BPA, while 2746 genes are affected by both phthalates and PCBs. This indicates that there is a similarity in the way these substances impair gene expression. Phthalates interact with a total of 6209 genes, with 342 unique interactions. They also have a small portion of interactions in common with PCBs, which altogether interact with 7821 genes. The intersection of the three EEDs encompasses a total of 2983 genes, indicating a notable core of genes that may have regulatory significance in the context of vascular disorders (Figure 2B) (Appendix A). The 938 genes that exclusively interact with PCBs exhibit a distinctive genetic profile that may be associated with the specific chemical composition and biological effects of PCBs. This particular subset of individuals may offer insights into the precise cardiovascular and vascular disorders linked to exposure to PCBs (Figure 2B) (Appendix A).

Based on the Venn diagram, our study has identified a set of genes implicated in both vascular remodeling and PAD. The diagram illustrates that there are 12,678 genes exclusively associated with PAD, and 1052 genes uniquely involved in vascular remodeling. There is also a significant overlap of 9724 genes between the two conditions, suggesting a shared genetic framework that may contribute to the pathophysiology of PAD and the process of vascular remodeling. This shared genetic landscape highlights the potential for targeted therapeutic strategies that could address the common molecular pathways in this vascular condition (Figure 1) (Appendix A).

### 2.4. Interactions among Estrogen, EEDs, and PAD Vascular Remodeling Genes

The Venn diagram presented in the figure offers a detailed representation of the gene interaction profiles among estrogen-interacting genes, EED-interacting Genes, and genes implicated in vascular remodeling in PAD. The analysis identified 1065 genes that interact with estrogen and a larger set of 2983 genes that interact with additional EEDs, indicating a diverse range of possible endocrine disruption within the genome. The intersection of genes shared between the estrogen-interacting and EED-interacting groups comprises 691 genes (Figure 3). This fraction may reflect a crucial convergence of regulatory networks impacted by both natural hormonal activity and exogenous endocrine disruptors. Moreover, there is a substantial quantity of genes (414) that are included in all three groups as shown in Figure 3 (Appendix A). This suggests that these genes play a crucial role in the development of PAD and the process of vascular remodeling. Additionally, these genes could be influenced by both estrogen and other EEDs.

The graphic also indicates a significant quantity of PAD-related vascular remodeling genes, totaling 9724. Out of the total number of genes (7009), the bulk of them do not overlap with estrogen or EED-interacting genes as shown in Figure 3 (Appendix A). This suggests the existence of distinct pathways that are exclusive to the disease process itself. The overlap between PAD-related vascular remodeling genes and EED-interacting genes, consisting of 1846 genes (Figure 3), highlights the potential influence of environmental disruptors on the course of the disease.

Furthermore, the analysis reveals a subset of 414 genes that are exclusively associated with the interaction between PAD-related vascular remodeling genes, EEDs, and estrogen. This suggests a focused estrogenic impact on the vascular remodeling component of PAD. The presence of 32 additional genes that are common to all three groups is highly significant, indicating a strong correlation with both the physiological and pathological processes influenced by hormonal and environmental variables in vascular remodeling.

This diagram illustrates the complicated network of gene interactions, offering detailed knowledge of how genes associated with the endocrine system may contribute to the various causes of PAD. It also prompts considerations regarding the potential cumulative impact of estrogen and EEDs on the process of vascular remodeling, emphasizing specific areas that require focused research and intervention in the treatment of PAD.

### 2.5. Differentially Expressed Genes in PAD

In our investigation of gene expression patterns associated with PAD, a common kind of systemic atherosclerosis that causes the narrowing of leg arteries over time, we used RNA sequencing data from the GSE27034 dataset to analyze gene expression patterns. Circulating monocytes, which establish and retain contact with the arterial wall, can serve as indications of vascular abnormalities in PAD conditions. In order to identify genes with modified expression patterns, we performed a differential gene expression analysis on peripheral blood mononuclear cells (PBMCs) obtained from patients with PAD and compared them to a control group of 18 individuals without PAD. Our research showed that many genes displayed significant changes in expression, with log fold changes (logFC) greater than 1 in magnitude, indicating either upregulation or downregulation in the context of PAD. Our differential expression analysis revealed a distinct set of genes significantly dysregulated in PAD (Appendix A). Specifically, we observed upregulation in genes such as C5orf28 and ZNF207. In contrast, several genes, including FCGR3A, EGR3, FFAR2, CXCL8, PTGS2, and G0S2, were notably downregulated as shown in Figure 4 (Appendix A). A volcano plot was constructed to effectively display these genes, highlighting the significant shifts in expression levels and the importance of their potential role in PAD pathophysiology.

### 2.6. Gene Network and Pathway Analysis of Interacting PAD-Related Vascular Remodeling Genes

Our gene analysis focused on 33 key genes out of a pool of 414 possible candidates to better understand the intricacies of PAD and its related vascular remodeling. The genes were chosen based on their significant correlations with the interactions between PAD-related vascular remodeling and EEDs (Appendix A). This was determined by analyzing data from Venn diagrams and identifying gene expression changes from the GEO database. In total, 1856 genes were found to have significant changes in expression with a *p*-value < 0.05. Using Ingenuity Pathway Analysis (IPA), we identified and analyzed many pathways. The results of our study revealed a notable increase in important biological processes such as epithelial–mesenchymal transition, hypoxia signaling in the cardiovascular system, and cardiac hypertrophy signaling as shown in Figure 5A (Appendix A).

Further exploring the topic, the module analysis conducted by IPA provided a clearer understanding of the complex network of gene interactions. It especially revealed the connections between certain genes and important molecular functions (FXs) and canonical pathways (CPs). For instance, the gene ID3 has been associated with various functional expressions (FXs) such as the movement of vascular endothelial cells, the ability of microvasculature to allow substances to pass through, and specific cellular CPs including PI3K/AKT signaling, atherosclerosis signaling, nitric oxide signaling in the cardiovascular system, and vascular endothelial growth factor (VEGF) signaling as shown in Figure 5B (Appendix A). Furthermore, FOS typically refers to a family of transcription factors known as Fos proteins linked to an FX that resulted in the excessive growth of heart muscle cells and the duration of smooth muscle cell life, while a CP was connected to endothelial nitric oxide synthase (eNOS) signaling. The association between PTP4A1 and the pathogenesis of coronary artery disease was observed to be substantial. Additional genes, including ID3, PTP4A1, LY6E, NAMPT, GADD45A, PDGF-BB, NFKB (complex), and Akt, were discovered as crucial elements in the network. Each of these genes is associated with a range of pathways and functions that are critical to vascular health and illness as shown in Figure 5B (Appendix A).

Our gene network and pathway analysis has provided a comprehensive understanding of the genetic interactions in PAD, revealing the complex network of gene interplays and the pathways they affect. These interactions are influenced by environmental factors, either worsening or modifying their effects. This research not only improves our comprehension of the complex molecular mechanisms involved in PAD and vascular remodeling but also highlights the influence of environmental disruptors on these processes.

## 3. Discussion

Our research findings add to the increasing evidence that environmental factors, particularly EEDs, have a notable impact on the development of vascular disorders, such as PAD. By utilizing IPA for network analysis, we have uncovered the widespread impact of EEDs on gene expression and interactions. This has provided valuable insights into the complex molecular processes involved in vascular remodeling in the setting of PAD.

An important finding from our investigation is the significant interaction between EEDs and genes that are already linked to the vascular remodeling processes associated with PAD. The network analysis demonstrates that these interactions are not only on the outskirts, but rather play a fundamental role in the pathways that control crucial elements of vascular biology, such as the functioning of endothelial cells, the proliferation of smooth muscle cells, and the inflammatory response. The conclusion is significant, stating that exposure to EEDs could enhance the risk or severity of PAD, not only through direct effects on these pathways but also by altering a wider network of gene regulation.

The DE analysis in our results section expands on these findings by illustrating the distinct gene expression patterns in individuals with PAD, providing a detailed perspective on how certain genes may react to exposure to EEDs [29]. The diversity in gene expression patterns among patients indicates that the effect of EEDs on vascular health is intricate and probably affected by a combination of hereditary and environmental variables. This highlights the imperative of employing individualized strategies in the management and treatment of PAD, considering the distinct genetic composition of individual patients.

Additionally, our findings prompt significant inquiries regarding the mechanism by which EEDs impact gene expression and interaction. Evidence indicates that EEDs can imitate or hinder the function of natural hormones. However, our research suggests that their impact goes beyond mere disruption of the endocrine system. The network analysis suggests that EEDs can influence gene expression indirectly by modifying epigenetic factors or by changing signal transduction pathways. Gaining comprehension of these pathways is essential for formulating therapeutic techniques that can alleviate the impact of EEDs on the vascular system. The integrative bioinformatics analysis identified candidate genes that may be direct targets of EEDs, potentially leading to the dysregulation of vascular endothelial architecture and, subsequently, PAD. ID3, PTP4A1, LY6E, NAMPT, GADD45A, PDGF-BB, NFKB (complex), and Akt candidate genes were identified through analysis.

The involvement of the NFκB signaling pathway in age-related vascular endothelial dysfunction holds substantial significance for human health. NFκB acts as a pivotal regulator of gene expression governing numerous aspects of the vascular system, including cell adhesion, inflammatory responses, and oxidative balance [30]. As humans age, endothelial function typically deteriorates, characterized by diminished endothelium-dependent dilation, heightened oxidative stress, and an inclination towards a pro-inflammatory state—all factors that contribute to the decline in vascular health. The activation of NFκB can be instigated by a variety of stimuli, such as inflammatory cytokines or mechanical stress exerted on the vascular walls, which in turn promotes transcriptional activities predisposing the vasculature to dysfunction and atherogenic changes [31].

Parallel to NFκB, Platelet-Derived Growth Factor-BB (PDGF-BB) plays a crucial role in vascular endothelial dysfunction and pulmonary arterial hypertension (PAH). It aids in the survival of pulmonary arterial endothelial cells (PAECs) under low-oxygen conditions, thereby promoting pulmonary vascular remodeling and the progression of PAH [30,32,33]. Furthermore, PDGF-BB influences pulmonary vascular tone by modulating signaling pathways involving cAMP, cGMP, NO, and actin polymerization, which affect the contractility of pulmonary vessels upon PDGF-BB stimulation [32]. These findings highlight the significance of PDGF-BB in endothelial function and vascular remodeling in PAH. The gene LY6E is emerging as a crucial contributor to vascular endothelial dysfunction through its role in a plethora of signaling pathways. Its participation in enhancing TGF-β signaling, immune evasion, and INF-γ signaling pathways has implications for tumorigenic advancement. LY6E’s interaction with the PI3K/Akt pathway [34], and its subsequent influence on HIF-1α transcription [35], outlines a mechanism pivotal to pathological processes such as tumor growth and angiogenesis. The expression levels of LY6E have been correlated with diminished survival rates across various cancers, accentuating its potential as both a prognostic indicator and a therapeutic target. Collectively, these insights paint a portrait of LY6E as a multifaceted gene implicated in the intricate ballet of immune responses, oncogenesis, and vascular development. Its impact on endothelial dysfunction and disease progression is of considerable interest, offering new vantage points from which to explore therapeutic strategies for vascular-related pathologies and cancer.

ID3 is integral to vascular function, influencing both the movement and integrity of endothelial cells, and is a component of pivotal signaling pathways such as PI3K/AKT, atherosclerosis, NO, and VEGF signaling [7,10]. FOS, conversely, contributes to the growth and longevity of cardiomyocytes and intersects with eNOS signaling, while PTP4A1 has a notable role in the development of coronary artery disease [36]. Investigations have established that Id proteins, notably Id3, are fundamental to cardiovascular health, impacting everything from heart formation to vascular disease, including atherosclerosis and pulmonary hypertension [36]. Id3 is especially recognized for fostering angiogenesis and managing endothelial progenitor cells [36]. The absence of Id genes can compromise vascular integrity, essential for the development of hematopoietic stem cells [37]. Furthermore, Id3 is implicated in the estrogen-induced signaling pathways that regulate vascular smooth muscle cells [38].

PTP4A1, a tyrosine phosphatase, is significant in endothelial cell dynamics and coronary artery disease. It is related to PTP4A3, which is known to foster endothelial cell movement by activating VEGF signaling; the absence of this phosphatase is associated with reduced cell migration [39]. The activation of PTP4A3 by VEGF is crucial for cellular movement and actin rearrangement in human endothelial cells [39]. Additionally, PTP4A1 is understood to counteract vascular inflammation via upstream stimulatory factor 1 (USF1) [40], underscoring its multifaceted role in regulating cell growth and migration, which are vital to vascular health and disease [40].

FOS is implicated in several cellular functions, including those associated with coronary artery disease and endothelial cell migration. The migratory activity of endothelial progenitor cells, inversely related to coronary artery disease risk, underscores the significance of endothelial migration in vascular health [41]. Shear stress from blood flow is a critical factor in the natural history of coronary artery disease, influencing the progression of atherosclerosis and vascular remodeling through the migration, differentiation, and proliferation of vascular smooth muscle cells [42]. Additionally, endothelial dysfunction is a critical player in atherosclerotic plaque development, contributing to structural plaque changes and increased vulnerability [43].

The interplay between NFκB, PDGF-BB, LY6E, ID3, PTP4A1, and FOS in age-related vascular endothelial dysfunction and related pathologies underscores the complexity of molecular mechanisms governing vascular health. NFκB’s role in inflammatory responses and oxidative balance, coupled with PDGF-BB’s involvement in pulmonary arterial hypertension and vascular remodeling, highlights the multifaceted nature of these factors in vascular dysfunction. Additionally, LY6E’s impact on signaling pathways and its implications in tumorigenesis and vascular development, along with ID3’s influence on endothelial cell dynamics and cardiovascular health, further exemplify the intricate regulatory networks at play. The involvement of PTP4A1 in endothelial cell migration and coronary artery disease, alongside FOS’s contribution to endothelial cell functions, underscores the significance of these genes in maintaining vascular integrity and function. Collectively, these findings provide valuable insights into the molecular underpinnings of vascular endothelial dysfunction and offer potential avenues for therapeutic intervention in vascular-related pathologies. Moreover, the endothelial dysfunction in diseases, such as rheumatoid arthritis and systemic lupus erythematosus, involves both innate and adaptive immune responses, including macrophage activation and autoreactive T-helper-1 lymphocyte proliferation, leading to atherosclerosis and obliterative vasculopathy [44].

Furthermore, this study’s ramifications transcend the biological domain and encompass the field of public health policy. Considering the extensive prevalence of EEDs in the environment and their potential influence on vascular health, our results emphasize the necessity of implementing regulatory actions to decrease EED exposure among the population. This is especially relevant for persons who are susceptible to PAD or those who already experience the condition.

As we further explore the consequences of our discoveries, we must also take into account the possibility of confounding variables in our research. For instance, the gene expression patterns reported in patients with PAD may be affected by simultaneous medical illnesses, lifestyle variables, or the use of medicines. In order to identify the precise effects of EEDs on gene expression and interactions in PAD, future studies must consider these characteristics.

Ultimately, our research offers a valuable perspective on the intricate connection between environmental disruptors and vascular disease. Understanding the gene networks and pathways that interact with EEDs provides a useful platform for future research focused on uncovering the complex causes of PAD. As we enhance our comprehension of these interactions, we create opportunities for targeted medical intervention and approach the objective of precision medicine in treating vascular disorders.

## 4. Materials and Methods

### 4.1. EED Selection and Data Integration

This study was conducted in three different steps: [1] selecting estrogens and EEDs to which exposure has an association with cardiovascular and vascular disease through previously published evidence via literature review; [2] identifying (a) common estrogen-interacting genes, (b) common EED (polychlorinated biphenyls (PCBs), bisphenol A (BPA), phthalates)-interacting genes, and (c) common vascular remodeling and peripheral arterial disease genes; [3] identifying and visualizing differentially expression genes in PAD; and [4] visualizing those particular gene networks and identifying their biological pathways and predicting candidate genes that can potentially mediate PAD outcomes via vascular remodeling. We used both the Comparative Toxicogenomic (CTD) and NCBI Gene Omnibus Expression (GEO) databases to support our study, as further discussed below. Figure 6 demonstrates the workflow of this study, and Table 1 shows the epidemiological and laboratory-based studies that show an association between EEDs and cardiovascular/vascular diseases.

Inclusion Criteria:

We included the data based on the following criteria: Research that offers epidemiological, laboratory, or clinical trial evidence establishing a connection between estrogens and EEDs (particularly PCB, BPA, and phthalates) and cardiovascular and vascular illnesses. Genes that have been recorded in the CTD interact with specific EEDs and estrogens in relation to cardiovascular and vascular illnesses.

Exclusion criteria:

We excluded the data or studies that do not particularly investigate the connection between estrogens or EEDs and cardiovascular/vascular disorders. We also excluded non-peer-reviewed studies or studies published in languages other than English. Additionally, we excluded data from the CTD that lacked evidence of gene interactions with selected EEDs and estrogens in the context of cardiovascular and vascular disorders.

### 4.2. Curating the Estrogen/EED-Interacting Genes, Vascular-Remodeling-Interacting Genes, and Peripheral Arterial Disease (PAD)-Interacting Genes

The CTD, an accessible online resource, plays a pivotal role in enhancing our understanding of the impact of environmental factors on human health. This database was utilized to pinpoint a set of specific genes, identified through careful curation. These genes were recognized for their dual role as both estrogenic and environmental disruptors, as documented in various scientific studies.

The selection of genes for this study was informed by their established links to cardiovascular and vascular diseases. This connection was supported by evidence from epidemiological studies, animal research, and human clinical data, as detailed in Table 1 of this study. The CTD facilitated the extraction of data on genes that interact with humans, focusing particularly on those genes that respond to certain environmental chemicals. In the CTD, a rigorous manual curation process is employed, where interactions between chemicals and genes in both vertebrates and invertebrates are meticulously documented based on published literature, primarily sourced from PubMed. For this study, genes that showed interactions with selected environmental chemicals (such as PCBs, BPA, and phthalates) as well as with synthetic estrogen diethylstilbestrol (DES) and the natural estrogen estradiol-17 beta (E2) were downloaded and subsequently analyzed for commonalities. A similar approach was adopted to identify genes related to vascular remodeling and PAD. The MyVenn tool within the CTD was instrumental in pinpointing specific genes that were common across different categories, thereby facilitating a more nuanced understanding of their roles in human health in relation to environmental exposures.

### 4.3. Differential Gene Expression Analysis

Data from the NCBI GEO database, a public functional genomic data repository that supports Minimum Information About a Microarray Experiment (MIAME)-compliant data submissions, were used to demonstrate how differentially expressed genes (DEGs) and gene network interactions play a crucial role in the interaction of EEDs in a PAD population. The results were supported by GSE27034 [20,28], which was downloaded from GEO and used as secondary data during our analysis. The original study information is listed in Table 2 below and consisted of 37 patient samples, 19 PAD and 18 control samples. Raw fastq sequence reads were initially processed to remove adapter sequences and low-quality data using Trim Galore [45], with adjustments made for paired-end reads. These processed reads were then aligned to the Hg38 human reference genome using RNA STAR, a tool designed for gapped alignments [46]. Gene expression quantification was performed with featureCounts, which provided counts for genomic features of each gene [46,47]. We utilized the Limma-Voom R package to identify differentially expressed (DE) mRNAs from the featureCounts output [48,49,50]. The analysis involved comparing mRNA expression between patient and control groups using functions such as “model.matrix”, “lmFit”, “eBays”, and “topTable”. Log2 fold change (log2FC) values for each mRNA were calculated by comparing the expression levels in patient samples to those in control samples. The threshold for statistical significance was set at *p* < 0.05.

### 4.4. Gene Set Enrichment and Network Analysis

The data were systematically reviewed to pinpoint significantly upregulated or downregulated DEGs. We delved deeper to ascertain the pathways most affected by these changes, their components, and the associated signaling networks [51]. The gene set enrichment analysis (GSEA) was conducted using Ingenuity Pathways Analysis (IPA) software (https://digitalinsights.qiagen.com/products-overview/discovery-insights-portfolio/analysis-and-visualization/qiagen-ipa/# accessed on 8 January 2024) and its knowledge database [52]. The IPA’s Core Analysis feature was utilized, which encompasses modules for canonical pathways, upstream regulators, causal networks, and molecular networks [52].

## 5. Conclusions

Our study conclusively demonstrates the significant impact of EEDs on the molecular dynamics of PAD and vascular remodeling. Through the utilization of bioinformatics techniques, we have successfully charted the gene networks and pathways that are affected by environmental disruptors. This has provided us with fresh perspectives on the development of vascular disorders. The considerable intersection of genes that interact with EED and those implicated in vascular remodeling in PAD implies that exposure to these disruptors may intensify the progression of the illness. Additionally, our methodology has identified possible biomarkers for PAD, which could play a crucial role in the timely identification and treatment of this condition.

It is essential to convert these discoveries into practical use in the medical field as we progress. This will require additional confirmation of the identified genes and pathways in bigger groups of people and different populations, as well as the creation of medicinal substances that may precisely modify these targets. Furthermore, our research underscores the significance of environmental well-being concerning long-term illnesses, underlining the necessity of implementing laws that restrict exposure to detrimental EEDs.

The key advantage of our paper is its thorough bioinformatic analysis and broad use of databases, which offer valuable insights into the molecular pathways that cause vascular endothelial dysfunction. Nevertheless, this study’s constraints encompass its dependence on in silico methodologies, which might not comprehensively encompass the intricacy of biological systems and the inherent predispositions in bioinformatic tools. It is imperative to conduct experimental validation in clinical settings to verify the relevance of our findings.

In essence, this work sets the stage for a new period in vascular medicine where genomic data and environmental health are combined to enhance clinical practices and policy decisions. We aim for the knowledge acquired here to support the worldwide endeavor to combat PAD and enhance the well-being of individuals impacted by this incapacitating ailment. The small sample size in the GEO study for DEG analysis might restrict the statistical power and generalizability of our findings, increasing the risk of type II errors and biases. This is due to our stringent inclusion and exclusion criteria, which helped us select the most relevant studies for our analysis despite the limited sample size. To address this issue and identify the most significant potential genes, we compared DEGs from the GEO study with the gene list analyzed by CTD. Based on our strict criteria, there are few studies that directly align with our research objectives. However, the selected studies provided a satisfactory number of populations for identifying genes related to EEDs and PAD. This yielded sufficient data for DEG analysis and gene network interactions, highlighting the role of EEDs in PAD. Our careful selection process ensures that our analysis is based on relevant and high-quality data, albeit from a smaller pool of studies.

## Figures and Tables

**Figure 1 ijms-25-04502-f001:**
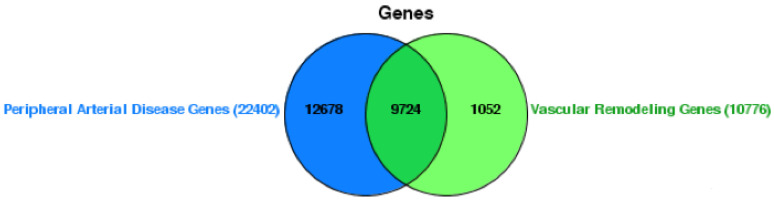
Venn diagram of gene overlap between peripheral arterial disease and vascular remodeling. This figure displays the overlap of genes associated with peripheral arterial disease (22,402 genes) and vascular remodeling (10,776 genes), with a substantial shared subset of 9724 genes, indicating potential common molecular pathways involved in both conditions.

**Figure 2 ijms-25-04502-f002:**
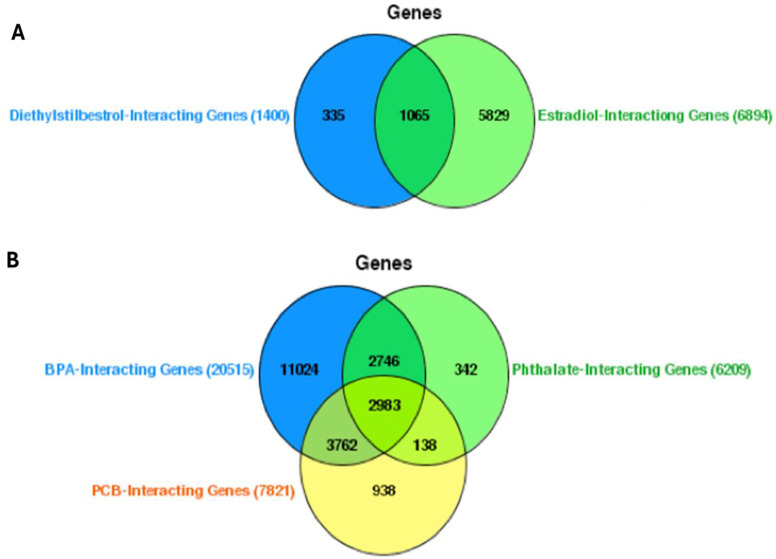
Gene interaction networks with EEDs and their overlaps. (**A**) Interactions of diethylstilbestrol (DES) and estradiol-17 beta (E2) with the genome. This part of the figure uses Venn diagrams to demonstrate the gene interactions specific to DES, shared with E2, and the broader genetic influence of E2 alone. (**B**) Composite interactions among bisphenol A (BPA), phthalates, and polychlorinated biphenyls (PCBs). The diagrams here reveal the individual and shared gene interactions with BPA, phthalates, and PCBs, indicating the intricate gene regulatory networks that are potentially significant in vascular disorders.

**Figure 3 ijms-25-04502-f003:**
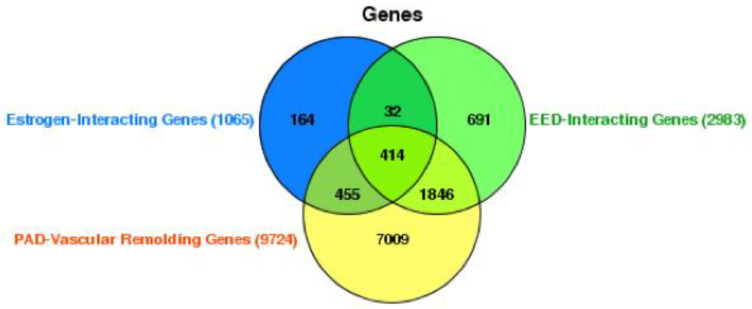
Gene interactions in peripheral arterial disease and endocrine disruption. This Venn diagram summarizes the overlap between genes interacting with estrogen, genes interacting with environmental estrogenic endocrine disruptors (EEDs), and genes involved in peripheral arterial disease (PAD)-related vascular remodeling. It identifies significant subsets of genes potentially central to both endocrine regulation and the pathophysiology of PAD.

**Figure 4 ijms-25-04502-f004:**
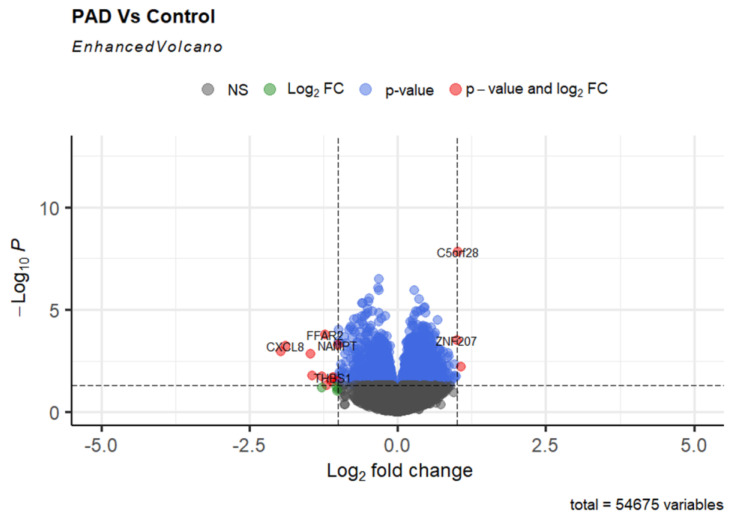
GEO data analysis. Volcano plot of gene expression in peripheral arterial disease (PAD). This plot visualizes significantly dysregulated genes in the PBMCs of PAD patients versus controls, highlighting upregulated genes like C5orf28 and ZNF207 and downregulated genes such as FCGR3A and EGR3. Each point represents a gene, with significant changes marked by their position relative to the fold-change and *p*-value thresholds.

**Figure 5 ijms-25-04502-f005:**
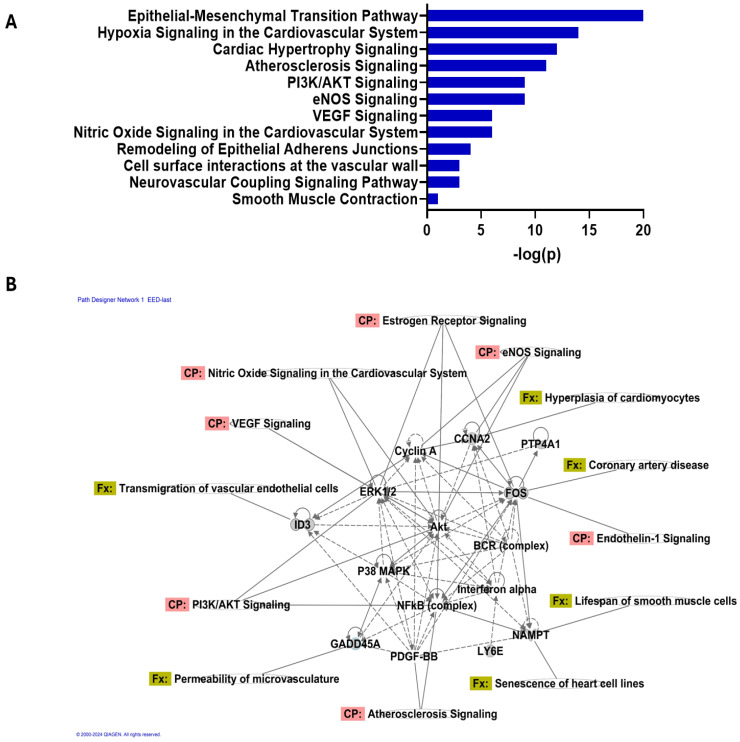
Integrated pathway and network analysis of peripheral arterial disease-associated gene interactions. (**A**) A bar plot illustrating the significant pathways identified by IPA in our study of 414 genes associated with peripheral arterial disease (PAD)-related vascular remodeling, environmental estrogenic endocrine disruptors (EEDs), and estrogen. The pathways are ranked by −log10 (*p*-value) indicating the level of upregulation in pathways. (**B**) The network presents the gene network analysis, revealing the intricate web of direct and secondary gene interactions and highlighting key nodes suggesting their central role in the molecular mechanisms of PAD-related vascular remodeling and the impact of EEDs on these processes.

**Figure 6 ijms-25-04502-f006:**
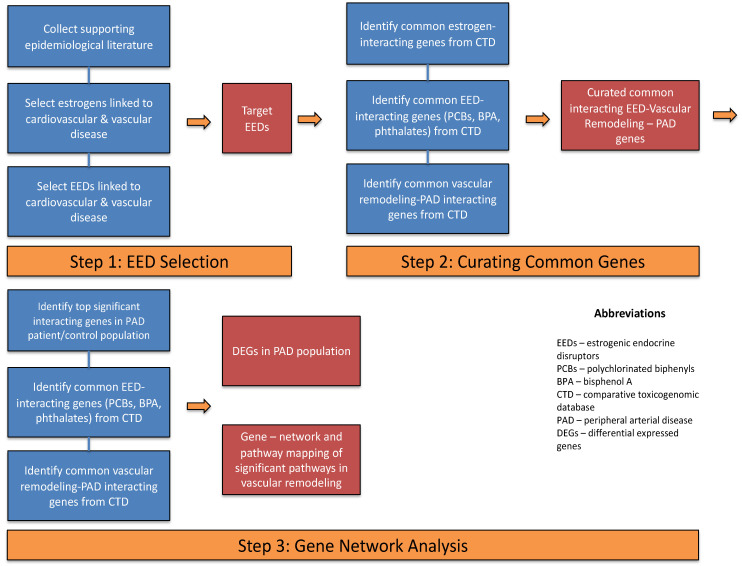
Schematic representation of the study workflow. The figure illustrates the comprehensive process undertaken in this research, encompassing a literature review for gene selection, curation of common interacting genes using CTD, and the subsequent gene network analysis through diverse methods, including gene network and pathway mapping.

**Table 1 ijms-25-04502-t001:** Epidemiological and laboratory studies showing associations of estrogen and EEDs (PCB, BPA, and phthalates) with cardiovascular and vascular diseases. Abbreviations: PCB—polychlorinated biphenyl, BPA—bisphenol A, PFAS—perfluoroalkyl.

Study Title	Author (Year)	Estrogenic Endocrine Disruptor
A Population-based prospective cohort study Bisphenol A, Hypertension, and cardiovascular diseases: Epidemiological, Laboratory, and Clinical Trial Evidence	Han and Hong (2016) [16]	BPA
Dietary exposure to polychlorinated biphenyls and risk of myocardial infarction in men—A population-based prospective cohort study	Bergkvist et al. (2016) [17]	PCB
Targeted basic research to highlight the role of estrogen and estrogen receptors in the cardiovascular system	Dworatzek and Mahmoodzadeh (2017) [18]	estrogen
Plastics and cardiovascular health: phthalates may disrupt heart rate variability and cardiovascular reactivity	Jaimes et al. (2017) [19]	phthalates
ID3, Estrogenic Chemicals, and the Pathogenesis of Tumor-Like Proliferative Vascular Lesions	Avecilla (2017) [20]	PCB, BPA, phthalates
The environmental pollutant, polychlorinated biphenyls, and cardiovascular disease: a potential target for antioxidant nanotherapeutics	Gupta et al. (2018) [21]	PCB
The Role of ID3 and PCB153 in the Hyperproliferation and Dysregulation of Lung Endothelial Cells	Doke (2018) [22]	PCB
Associations of Perfluoroalkyl and Polyfluoroalkyl Substances with Incident Diabetes and Microvascular Disease	Cardenas et al. (2019) [23]	PFAS
Dietary exposure to polychlorinated biphenyls and risk of heart failure—A population-based prospective cohort study	Åkesson et al. (2019) [24]	PCB
Environmental Contaminants Acting as Endocrine Disruptors Modulate Atherogenic Processes: New Risk Factors for Cardiovascular Diseases in Women?	Migliaccio et al. (2021) [25]	BPA, cadmium
Evaluation of Early Biomarkers of Atherosclerosis Associated with Polychlorinated Biphenyl Exposure: An in Vitro and in Vivo Study	Yang et al. (2022) [26]	PCB
Exposure to the Dioxin-like Pollutant PCB 126 Afflicts Coronary Endothelial Cells via Increasing 4-Hydroxy-2 Nonenal: A Role for Aldehyde Dehydrogenase 2	Roy et al. (2022) [27]	PCB

**Table 2 ijms-25-04502-t002:** Information of dataset provided by NCBI GEO. PAD was defined as an ankle–brachial index (ABI) ≤ 0.9 (N = 19), while controls had an ABI > 1.0 (N = 18) (Masud et al., 2012 [28]).

NCBI GEO Database Information	
Accession Number	GSE27034
Experiment Type	Expression Profiling by Array
Peripheral Arterial Disease Samples	19
Control Samples	18
Citation	Masud et al., 2012 [28]

## Data Availability

Data are contained within the article and Appendix A.

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
