# Peer review of "Integrative Bioinformatics–Gene Network Approach Reveals Linkage between Estrogenic Endocrine Disruptors and Vascular Remodeling in Peripheral Arterial Disease"

_ijms, 2024, doi:10.3390/ijms25084502_

Round 1

Reviewer 1 Report

Comments and Suggestions for Authors

Impressive review. Following are my few suggestions to improve the manuscript.

References, and their citations throughout the manuscript, must be reported according to the IJMS’s Instructions for Authors (>Reference List and Citations Guide> MDPI Reference List and Citations Style Guide or Chicago Reference List and Citations Style Guide). 

Introduction, Lines 41-44, “…PAD also acts as a standalone predictor of unfavorable cardiovascular events. Given its high incidence, which affects around 202 million 42 people worldwide, PAD is linked to significant morbidity, mortality, and healthcare expenditures.”: this is a crucial point when dealing about PAD. A very recent review focuses on early detection of PAD, as fundamental to reduce the risk of major adverse cardiovascular and limb events. Please, cite it [Martelli, E.; Enea, I.; Zamboni, M.; Federici, M.; Bracale, U.M.; Sangiorgi, G.; Martelli, A.R.; Messina, T.; Settembrini, A.M. Focus on the Most Common Paucisymptomatic Vasculopathic Population, from Diagnosis to Secondary Prevention of Complications. Diagnostics (Basel)202313:2356-2376. doi: 10.3390/diagnostics13142356].

Methods, section 2.1, line 127, “… (PCB, BPA, …”: explain the meaning of these acronyms here, the first time you report them, and not later (see Methods, section 2.2, line 174-5) (see Results, section 3.2, lines 236-7).

Methods, section 2.2, line 161, “The Comparative Toxicogenomic Database (CTD) …”: use just the acronym, its meaning has been already explained before.

The meaning of each acronym should be reported in the footnote of each figure: correct figures 2, 4, and 6. 

Results, section 3.3, line 267, “… as seen in the accompanying figure.”: better to say “… as seen in figure 3.”.

Results, section 3.3, line 271-7, “Among these, 335 genes are exclusively associated with Diethylstilbestrol, while 1,065 genes overlap with those that also interact with Estradiol-17 beta. In total, Estradiol-17 beta interacts with 6,894 genes (Figure 1A) (Supplementary file 1). The presence of an overlap between Diethylstilbestrol and Estradiol-17 beta suggests that they share a route or have comparable methods of action. This is supported by the fact that 5,829 genes do not interact with Diethylstilbestrol. This indicates that Estradiol-17 beta has a wider range of influence on the genetic landscape (Figure 1A).”: be consistent, use the acronyms DES and E2 already introduced before (see Methods, section 2.2, lines 175-6).

Results, section 3.6, line 366, “Using Ingenuity Pathway Analysis (IPA) …”: use just the acronym, its meaning has been already explained before.

Results, section 3.6, line 376, “… and VEGF Signaling …”: explain the meaning of this acronym.

Results, section 3.6, line 379-80, “Furthermore, FOS was linked …”: explain the meaning of this acronym.

Results, section 3.6, line 381, “… was connected to eNOS Signaling …”: explain the meaning of this acronym.

Discussion, line 470, “… atherosclerosis, nitric oxide, and VEGF signaling…”: you have introduced the acronym NO before, use it.

In the manuscript there are many other not explained acronyms (above all for genes). To facilitate the reader, I suggest the authors to create a Glossary at the beginning of the manuscript, reporting the meaning of each acronym used.

Author Response

Dear Reviewer,

 Thank you for providing constructive feedback on our manuscript. To assist the reviewers in evaluating the revisions, we have highlighted the changes made in response to Reviewer 1's comments in blue.

Reviewer 2 Report

Comments and Suggestions for Authors

The paper is interesting and well written. The authors investigated the profound effect of estrogenic endocrine disruptors n pivotal genes such as
ID3, LY6E, FOS, PTP4A1, NAMPT, GADD45A, PDGF-BB, and NFKB all of which play significant roles in  peripheral arterial disease pathophysiology. The study enhanced the understanding of genomic alterations induced by estrogenic endocrine disruptors in vascular remodeling processes. The methodology and statistical analysis is adequate and coerent with the endpoints of the study. The results are well described and the discussion is adequate with the objectives. I suggest to discuss if these genes may impact on accelerated atherosclerosis in patients with chronic immune-mediated diseases particularly on the role of VEGF in endothelial dysfunction ( see nad add as reference paper by Murdaca et al concerning endothelial dysfunction in autoimmunity).

Comments on the Quality of English Language

Minor english editing

Author Response

Dear Reviewer,

 Thank you for providing constructive feedback on our manuscript. To assist the reviewers in evaluating the revisions, we have highlighted the changes made in response to Reviewer 2's comments in red.
